# Sovereign Credit Spread Spillovers in Asia

**Biao Guo [1], Qian Han [2], Jufang Liang [3], Doojin Ryu [4],\*  and Jinyoung Yu [4]**

1    School of Finance, Renmin University of China, Beijing 100872, China; biao.guo@outlook.com
2    Wang Yanan Institute for Studies in Economics, Xiamen University, Xiamen 361005, China; hanqian@gmail.com
3    College of Finance and Statistics, Hunan University, Changsha 410006, China; ljufang@hnu.edu.cn
4    College of Economics, Sungkyunkwan University, Seoul 03063, Korea; mydkfkqldk@naver.com
\*    Correspondence: doojin.ryu@gmail.com

**Abstract:** Sovereign credit default swap (CDS) spreads exhibit strong co-movements across Asian countries and regions, including both emerging and developed economies. After controlling for global impacts, we examine the regional lead-lag relationships among changes in ten Asian sovereign CDS spreads. We use the pairwise Granger causality test to find that lagged changes in Kazakhstan's sovereign CDS spreads significantly predict changes in other Asian sovereign CDS spreads. By estimating the news-diffusion model, we find evidence that this predictive relationship may be explained by information diffusion. Furthermore, we find that lagged changes in Kazakhstan's CDS spreads have significant out-of-sample predictive power for other Asian economies, providing practical implications for sustainable investments and risk management.

**Keywords:** Asian market; emerging financial market; sovereign contagion; sovereign credit default swap; sustainable investments

## 1. Introduction

A credit default swap (CDS) is a type of protection guarantee or contract where its buyer provides a payment to its seller, which is commonly known as the CDS spread. In return, the buyer is compensated in the case of an actual default (Bhanot and Guo, 2012 [1]; Guo, 2016 [2]; Guo and Newton, 2013 [3]; Guo et al., 2020 [4]; Park et al., 2019 [5]). Some studies suggest domestic determinants of sovereign CDS spreads. For example, Bouri et al. (2017 [6]) investigated six frontier economies (including Croatia and Kazakhstan) and seventeen developing economies (including Brazil and Chile) and found that the volatility of commodity sectors, especially energy commodities, significantly affects a country's CDS spread. Nevertheless, most changes in sovereign CDS spreads can be explained by global factors, as has been clearly shown in numerous studies. For example, Pan and Singleton (2008 [7]) applied a single latent factor model to the full-term structures of CDS spreads with maturities of one, two, three, five, and ten years for Mexico, Turkey, and Korea. They found that the credit spreads in these three countries are strongly related to the Volatility Index (VIX) of the Chicago Board Options Exchange (CBOE). Longstaff et al. (2011 [8]) analyzed the CDS dataset from multiple countries. They claimed that U.S. macroeconomic factors (i.e., global factors) affect sovereign credit spreads more than local factors do in each country. Li et al. (2020 [9]) investigated volatility spillovers between Brent oil prices, the VIX, and the sovereign CDS spreads of some oil-importing (South Africa and Turkey) and exporting countries (Kazakhstan and Russia) and Kazakhstan's role among them. They found that Kazakhstan's five-year sovereign CDS can be utilized for portfolio diversification since it exhibits a significant correlation with those of other countries, and they further proposed a potential expansion of this research by analyzing the determinants of CDS spreads. Other influential studies on sovereign contagion effects include those by Ang and Longstaff (2013 [10]), Arora and Cerisola (2001 [11]), Benzoni

et al. (2014 [12]), Dailami et al. (2008 [13]), Gande and Parsley (2005 [14]), Geyer et al. (2004 [15]), Mauro et al. (2002 [16]), Remolona et al. (2008 [17]), and Weigel and Gemmill (2006 [18]).

In addition to global effects, researchers have provided evidence of regional sovereign contagion risks. Guo et al. (2020 [19]) found that local components account for 16% of CDS variability, on average, across 37 countries, in addition to the 52% of CDS variability accounted for by global components. Kalbaska and Gatkowski (2012 [20]) and Manasse and Zavalloni (2013 [21]) demonstrated the existence of sovereign contagion among the CDS markets in Europe, and Caceres and Unsal (2013 [22]) found sovereign contagion risks in Asia.

Less well explored are the lead-lag relationships among countries within a region after removing global influences. We contribute to this literature by examining sovereign risk spillover effects in Asia using ten daily sovereign CDS spreads from February 2011 to December 2014. We discover that Kazakhstan plays a leading role among Asian economies in the sovereign CDS market. Specifically, we demonstrate that lagged changes in Kazakhstan's CDS spreads are significant predictors of changes in other Asian sovereign CDS spreads. In addition, we show that adding changes in Kazakhstan's spreads delivers statistically significant out-of-sample gains for most Asian economies. These results together indicate Kazakhstan's leading role in the Asian sovereign CDS market.

When investigating the lead-lag relationships between countries, it is important to note that this study mainly focuses on the relationships among emerging economies. Previous studies on similar topics mostly concentrated on advanced economies or the relationship between developed and developing economies. This study uniquely investigates CDS spread spillovers among Asian emerging markets since the markets provide a different research context. Many previous studies emphasized the importance of economic analyses of emerging economies and the uniqueness of their market structures, differing from those of advanced economies. Alon and Rottig (2013 [23]) suggested that emerging markets are different from their advanced counterparts in that they possess different types of political and economic structures and cultural conventions. They also suggested that different economic contexts of these developing economies provide new environments to study conventional ideas or theories, leading to further generalization or adjustment of traditional ones. Aizenman et al. (2016 [24]) asserted that emerging markets exhibit much faster growth compared to advanced markets', possibly implying the two-speed world economy theory. Mendy and Rahman (2019 [25]) studied the internationalization of small to medium-sized enterprises (SMEs) in an emerging market, considering potential barriers regarding 'people-relevant' (e.g., the language difference, scarcity of experienced workers, etc.) and 'institution-relevant' factors (e.g., how unstable political activities are, how cumbersome the legal procedures for internationalization are, etc.). They asserted that emerging markets are vastly different from advanced ones, particularly regarding their people and government-related settings.

However, emerging markets are not completely detached from the world economy. Instead, they rather exhibit significantly interactive, and yet autonomous, behaviors in response to the sovereign and financial activities of advanced markets. A number of studies illustrated the connection between developed and emerging markets. For example, Bowman et al. (2015 [26]) revealed that U.S. monetary policies significantly influence the yields of sovereign bonds in emerging markets. Similarly, Fink and Schüler (2015 [27]) found that shocks upon U.S. financial factors, such as its gross domestic products and consumer price index, play as significant determinants of macroeconomic dynamics in emerging markets. In addition, Song et al. (2018 [28]) investigated the dynamic relationship between advanced and emerging markets and found that financial variables, especially U.S. S&P 500 returns and the VIX, explain this relationship better than macroeconomic variables, including the exchange rate and interest rates. At the same time, there are studies that address independent aspects of emerging markets, specifically regarding their resilience to the international economic status. Aizenman et al. (2016 [24]) claimed that emerging markets displayed greater resilience during the global financial crisis period around 2008, particularly in comparison with advanced economies such as the U.S. and the Eurozone. Additionally, Rahman and Mendy (2019 [29]) investigated SMEs in an emerging market and

found that their internationalization is significantly affected by resilience-related factors, especially the barrier owing to language discrepancy. Thus, in line with the literature, we believe that it is academically necessary and timely to investigate the relationships among CDS spreads within the context of emerging markets.

Specifically, we first estimate a benchmark prediction model in which the only determinants of changes in sovereign CDS spreads are global factors. Similar to Pan and Singleton (2008 [7]), we approximate global factors using two variables representing the U.S. financial market: the CBOE VIX and S&P 500 returns. The VIX provides an implied volatility measure for S&P 500 index options and often represent the volatility of the U.S. financial market. The S&P 500 return represents the business and economic condition of the overall U.S. market and is often used as a determinant of changes in credit spreads, as suggested by Collin-Dufresne et al. (2001 [30]) and Guo et al. (2015 [31]). We find that the coefficient of the S&P 500 return is significantly negative at the 1% level for changes in nine of the ten sovereign CDS spreads, with Pakistan being the only exception. The VIX is a less significant determinant of Asian sovereign risk. Together, these two variables explain 15.27% of sovereign spread changes in our sample.

Second, we conduct standard pairwise Granger causality tests for lead-lag relationships in Asian CDS markets. The prediction model for economy *i* includes lagged changes in CDS spreads of its own and another country's in addition to the two U.S. variables. Because a market's sovereign risk is determined by global, regional, and domestic factors, we control for the U.S. variables to examine only the influences of regional and domestic factors, which allows us to concentrate on regional lead-lag relationships. Adding a country's own lagged spread changes removes autoregression in the time series and avoids spurious regression results. We find that changes in Kazakhstan's CDS spreads Granger-cause the CDS spread changes in eight of the nine Asian markets in this study. In comparison, changes in Chinese CDS spreads only Granger-cause changes in three other markets.

Third, we estimate a news-diffusion model, as in Rapach et al. (2013 [32]), to better understand the causes of Kazakhstan's leading role within the Asian sovereign CDS markets. Our generalized method of moment (GMM) estimation indicates that shocks from Kazakhstan's spread have statistically and economically significant effects on other sovereign CDSs. Moreover, information frictions explain a significant percentage of the predictive power of lagged changes in Kazakhstan's spreads.

Fourth, we estimate the out-of-sample predictability of lagged changes in Kazakhstan's sovereign CDS spreads. These changes deliver statistically significant out-of-sample improvements for the majority of Asian CDS markets relative to a historical average prediction model.

In sum, our results provide evidence that Kazakhstan plays a leading regional role in the Asian sovereign CDS market after controlling for global impacts from the U.S. The lead-lag relationships are both statistically significant and economically meaningful for predicting changes in other Asian sovereign spreads. This study is most closely related to that of Rapach et al. (2013 [32]) in that we adopt their test methodology. The main difference between our study and theirs lies in the financial markets examined. Rapach et al. (2013 [32]) examined international stock markets, whereas we focus on Asian sovereign CDS markets.

The CDS markets were introduced in the early 1990s, and the market started to grow rapidly in 2003 (https://www.isda.org). The credit derivatives are important trading vehicles for hedging, speculative, and arbitrage trading purposes of investors, but they have received little attention in the literature. This difference leads to an imbalance in research design, and thus, the findings of this study provide insights for international credit derivative traders, quantitative analysts, and portfolio managers.

Numerous economic reasons could explain why changes in Kazakhstan's sovereign CDS spreads may lead to changes in other Asian sovereign CDS spreads. One possible explanation is as follows. According to sovereign credit ratings by Standard and Poor's for the sample period of this study, from 2011 to 2014, Kazakhstan's rating is affirmed at 'BBB+,' which implies that it attracts more investor attention than the junk-rated (such as that of Pakistan, which is affirmed at 'B-' credit rating) or high-quality sovereign CDSs (such as those of China and Japan, both of which are affirmed at 'AA-' credit ratings) do. The variation in CDS spreads may be less meaningful for those with very high or very low ratings, as the credit risk characteristics of both types seldom change. Another explanation for Kazakhstan's leading role could be its location. Kazakhstan is a contiguous transcontinental country in Central Asia with some parts of its land in Europe, meaning that it has different trade partners than the other Asian economies in our sample. This unique feature may attract some European CDS market investors, and, thus, its sovereign CDS spreads may contain some information that is not captured by those in other Asian markets. Another possible reason may be that Kazakhstan historically has had an odd CDS since it has been traded for a long time with no actual underlying deliverable obligation (because Kazakhstan had no USD-based bond). Instead, it was traded on the assumption that Kazakhstan would issue the debt eventually, which it finally did so in October 2014. Before then, the bond that was used as the reference spread was of the state-owned oil and gas company Kazmunaygas. These bonds are traded widely enough to take into account the quasi-sovereign risk as opposed to the pure sovereign risk on the CDS. Thus, Kazakhstan's CDS spreads contain additional information about oil price uncertainties. Sharma and Thuraisamy (2013 [33]) found that oil price shocks can predict out-of-sample changes in six Asian CDS spreads. Exploring these alternative explanations is beyond the scope of our study, and we leave that analysis for future work.

The rest of this paper proceeds as follows. Section 2 describes and summarizes the data. Section 3 reports the results of benchmark predictive regressions. Section 4 presents the pairwise lead-lag relationships of the Granger causality tests. Section 5 demonstrates the estimation of the news-diffusion model. Section 6 explains the out-of-sample performance, and Section 7 concludes the paper.

## 2. Data Description

Our sovereign CDS spread data span the period from 14 February 2011 to 25 November 2014. The length of this sample period is mainly determined by the requirement that all economies considered must have reasonable variation in the data observations during the sample period. The final sample includes ten major Asian countries and regions: China (CHN), Indonesia (INA), Japan (JPN), Malaysia (MAS), Pakistan (PAK), the Philippines (PHI), South Korea (KOR), Thailand (THA), Kazakhstan (KAZ), and Hong Kong (HK). All except Pakistan are listed in the Markit iTraxx SovX Indices composed of liquid global sovereign entities. Table 1 reports major statistics for daily log changes in their CDS spreads, and Figure 1 presents daily CDS spreads for China, Hong Kong, India, Japan, Malaysia, Pakistan, the Philippines, South Korea, Thailand, Kazakhstan, the S&P 500 index, and the CBOE VIX index from 2011 to 2014. According Table 1, all have positive skewness and large kurtosis values, suggesting that changes in CDS spreads follow right-skewed and heavy-tailed distributions. Thailand and Kazakhstan have the smallest standard deviations among the group. CDS spread changes in the remaining countries and regions seem to be very volatile. CDS spread changes in the Philippines are more centered around the mean than those in other markets are. Pakistan has the largest kurtosis (24.3) and the highest standard deviation (3.29). Indeed, as shown in Figure 1, Pakistan's CDS spreads, which has the highest value in the group owing to the country's low credit rating, has decreased dramatically since early 2014. On the other hand, Kazakhstan's CDS spreads have the second-lowest kurtosis and volatility, suggesting that, although they are at high levels, they have been smooth relative to other spreads.

**Table 1.** Summary statistics of the sample.

|  | **Mean** | **Standard Deviation** | **Skewness** | **Kurtosis** | **Max** | **Min** |
|---|---|---|---|---|---|---|
|  | | Changes in CDS Spreads | | | | |
| China | 0.0055 | 3.2639 | 0.5991 | 7.0315 | 22.0664 | −11.4535 |
| Hong Kong | 0.0052 | 3.0661 | 0.3791 | 14.6453 | 19.3371 | −20.2941 |
| Indonesia | −0.0129 | 3.1754 | 0.5491 | 8.8149 | 20.2941 | −12.0231 |
| Japan | −0.0311 | 2.9189 | 0.6645 | 14.4397 | 25.6238 | −15.4206 |
| Malaysia | 0.0056 | 3.0126 | 0.4958 | 9.2253 | 18.4571 | −19.6665 |
| Pakistan | −0.0450 | 3.2880 | 0.3810 | 24.2555 | 30.8535 | −16.9700 |
| Philippines | −0.0514 | 3.0733 | 0.1881 | 9.1671 | 18.1411 | −16.5653 |
| Korea | −0.0732 | 3.1853 | 0.3472 | 6.6867 | 17.5863 | −13.4236 |
| Thailand | −0.0332 | 2.7154 | 0.3349 | 10.4432 | 15.2100 | −19.0001 |
| Kazakhstan | 0.0081 | 2.7592 | 0.3944 | 7.4724 | 16.8725 | −14.1781 |
|  | | Control Variables | | | | |
| S&P500 | 0.0461 | 0.9859 | −0.5905 | 8.6852 | 4.6317 | −6.8958 |
| VIX | 17.6623 | 6.2548 | 2.0109 | 7.0764 | 48.0000 | 10.3200 |

Notes*:* This table reports summary statistics for daily changes in sovereign CDS spreads, log-returns of the S&P 500 index, and the VIX from 14 February 2011 to 25 November 2014.

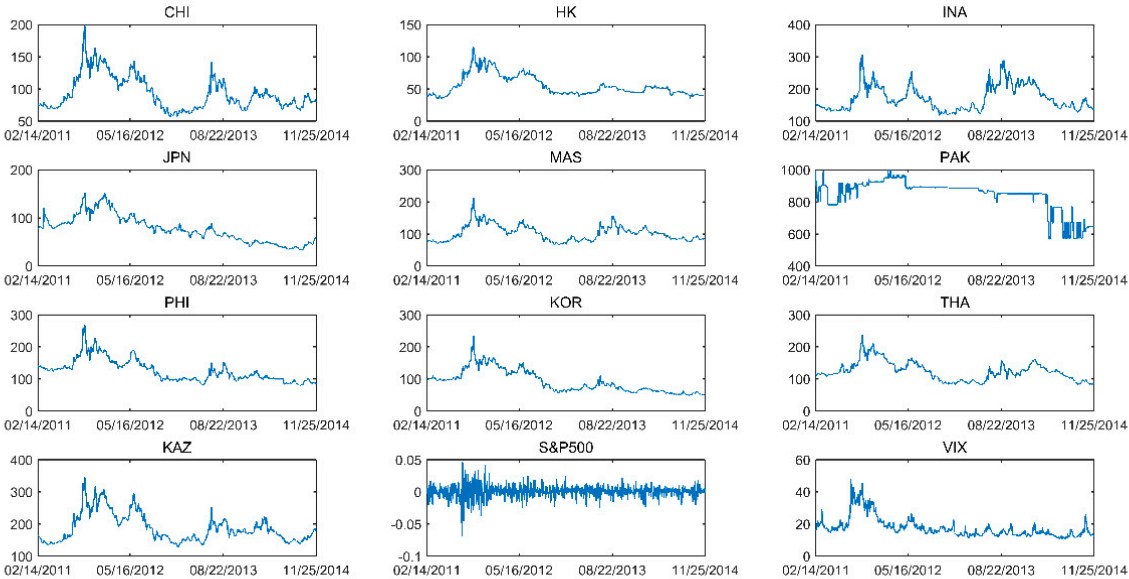

**Figure 1.** Asian sovereign CDS spreads and U.S. financial factors.

## 3. Predictive Regression Models

As Longstaff et al. (2011 [8]) point out, global factors make the most significant contribution to the daily variations in individual sovereign CDS spread changes. Hence, in the following regression models, we use two global factors: U.S. stock market performance, as proxied by S&P 500 index returns, and the CBOE VIX. We also examine other variables, such as the spread between the rate of return on six-month Treasury bills and the return on ten-year BB-rated industrial corporate bonds in the U.S. to reflect the default premium in the U.S. market, and find similar results. By controlling for economic determinants from the U.S., this benchmark model setup also allows us to focus on lead-lag relationships among Asian sovereign CDS spreads:

$$r_{i,t+1} = \beta_{i,0} + \beta_{i,S} r_{S,t} + \beta_{i,V} VIX_t + \varepsilon_{i,t+1}, \tag{1}$$

where $r_{i,t+1}$ is the daily log change in country $i$'s CDS spread, $r_{S,t}$ is the log return of the S&P 500 index, and $VIX_t$ is the level of the CBOE VIX. The estimation results are reported in Table 2. Values in parentheses in the second and third columns are heteroskedasticity-robust $t$-statistics for the tests of

$H_0$: $\beta_{i,S} = 0$ against $H_A$: $\beta_{i,S} < 0$ and $H_0$: $\beta_{i,V} = 0$ against $H_A$: $\beta_{i,V} > 0$, respectively. Following Rapach et al. (2013 [32]) and the related literature, we compute the significance of these *t*-statistics based on empirical *p*-values using a wild bootstrap procedure.

**Table 2.** Benchmark predictive regression model estimation results.

|  | $\beta_{i,S}$ | $\beta_{i,V}$ | $R^2$ |
|---|---|---|---|
| CHN | −1.5492 *** | 0.0067 | 22.102 *** |
|  | (−11.6897) | (0.3128) | (172.0837) |
| HK | −1.1702 *** | 0.0272 | 15.1194 *** |
|  | (−7.0242) | (0.9845) | (59.4243) |
| INA | −1.4721 *** | 0.0132 ** | 23.6955 *** |
|  | (−11.9168) | (0.6436) | (156.1066) |
| JPN | −1.1373 *** | 0.0181 | 15.3707 *** |
|  | (−8.9036) | (0.9592) | (114.8775) |
| MAS | −1.4721 *** | 0.0132 | 23.6955 *** |
|  | (−11.9168) | (0.6436) | (156.1066) |
| PAK | −0.0079 | 0.0128 | 0.0624 |
|  | (−0.093) | (0.9207) | (0.9410) |
| PHI | −1.4537 *** | 0.0103 | 22.1152 *** |
|  | (−11.254) | (0.5033) | (157.3976) |
| KOR | −1.6296 *** | 0.0073 | 25.6922 *** |
|  | (−12.4913) | (0.3211) | (190.7774) |
| THA | −1.2001 *** | 0.0167 | 19.6558 *** |
|  | (−11.483) | (0.9287) | (162.7776) |
| KAZ | −0.8163 *** | 0.0026 | 8.5644 *** |
|  | (−7.45) | (0.135) | (56.7986) |
| Pooled | −1.1934 *** | 0.0125 | 15.2674 *** |
|  | (−13.2758) | (0.8328) | (246.841) |

Note: This table reports ordinary least squares (OLS) estimates of $\beta_{i,S}$ and $\beta_{i,V}$ (denoted $\hat{\beta}_{i,S}$ and $\hat{\beta}_{i,V}$) and $R^2$ statistics for the predictive regression model: $r_{i,t+1} = \beta_{i,0} + \beta_{i,S} r_{S,t} + \beta_{i,V} VIX_t + \varepsilon_{i,t+1}$, where $r_{i,t+1}$ is the daily log change in country *i*'s CDS spread, $r_{S,t}$ is the log return of the S&P 500 index, and $VIX_t$ is the level of the CBOE VIX. The figures in parentheses in the second and third columns are heteroskedasticity-robust *t*-statistics; the *t*-statistics for $\hat{\beta}_{i,S}$ ($\hat{\beta}_{i,V}$) are used to test $H_0$:$\beta_{i,S} = 0$ against $H_A$:$\beta_{i,S} < 0$ ($H_0$:$\beta_{i,V} = 0$ against $H_A$:$\beta_{i,V} > 0$). Figures in parentheses below the $R^2$ statistics report heteroskedasticity-robust $\chi^2$ statistics for testing $H_0$:$\beta_{i,S} = \beta_{i,V} = 0$. The "Pooled" estimates impose the restrictions that $\beta_{i,S} = \bar{\beta}_S$ and $\beta_{i,V} = \bar{\beta}_V$ for all *i*. *, **, and *** denote significance at the 10%, 5%, and 1% levels, respectively, according to wild bootstrapped *p*-values.

With the exception of Pakistan, all the coefficients of the U.S. stock index return in Table 2 are significantly negative, consistent with the literature finding that U.S. stock market performance contributes significantly to changes in individual sovereign CDS spreads. The coefficients for the CBOE VIX are insignificantly positive for all markets except Indonesia.

Table 2 also reports $R^2$ values and heteroskedasticity-robust $\chi^2$ statistics to test the null hypothesis $H_0$: $\beta_{i,S} = \beta_{i,V} = 0$. For markets other than Pakistan and Kazakhstan, the $R^2$ value is around 23% and significant at the 1% level, suggesting that U.S. stock market returns, a global factor, have good explanatory power for these CDS spread changes. In comparison, U.S. stock market returns only explain about 8.5% of the total variation in Kazakhstan CDS spread changes, implying the existence of some other important factors that are left out of the regression. Global factors have no explanatory power at all for Pakistan, possibly because of the high illiquidity of Pakistan's CDS contracts, as shown in Figure 1.

Table 2 also includes the pooled estimates for regressions that impose the restrictions that $\beta_{i,S} = \bar{\beta}_S$ and $\beta_{i,V} = \bar{\beta}_V$ for all *i*. As in Rapach et al. (2013 [32]), the *t*-statistics for $\hat{\beta}_{i,S}$ and $\hat{\beta}_{i,V}$ are based on standard errors computed using a GMM procedure. Again, the stock return factor has negative and significant effects, but the VIX factor does not. The $R^2$ value decreases to 15% owing to the inclusion of Kazakhstan and Pakistan, but it is still quite significant.

## 4. Predictive Power of Sovereign CDS Spread Changes

In this section, we examine the predictive power of sovereign CDS spread changes through a Granger causality approach. Pairwise tests are carried out as follows:

$$r_{i,t+1} = \beta_{i,0} + \beta_{i,i}r_{i,t} + \beta_{i,j}r_{j,t} + \beta_{i,S}r_{S,t} + \beta_{i,V}VIX_t + \varepsilon_{i,t+1}, \tag{2}$$

where $r_{j,t}$ is the lagged term of country *j*'s CDS spread changes. Unlike the benchmark regression model in the previous section, this specification includes a country *i*'s lagged CDS spread changes in addition to controlling for the global factors of U.S. stock market returns and the VIX. As noted by Rapach et al. (2013 [32]) and other studies, it is important to incorporate $r_{i,t}$ as an explanatory variable to avoid the spurious test problem.

The OLS estimates of $\hat{\beta}_{i,j}$ and their heteroskedasticity-robust *t*-statistics are shown in Table 3. It is strikingly clear that lagged changes in Kazakhstan's CDS spreads significantly predict changes in the CDS spreads of all countries and regions in our sample except for Pakistan. The spread changes of China, India, Malaysia, and South Korea are only predicted by lagged changes in Kazakhstan's CDS spreads. The $\hat{\beta}_{i,KAZ}$ coefficients are all positive, indicating that increases in Kazakhstan's CDS prices tend to be followed by concern about default risk in other Asian economies. In addition, the sizes of these coefficients are much larger than those for the other countries and regions. On average, Kazakhstan's impact on other economies, 0.2253, is three (Malaysia) to one hundred (Japan) times larger than those of other economies. All of this evidence points to the Kazakhstan CDS market's leading role in perceptions of Asian markets' default risks.

It is worth noting that changes in Thailand's CDS spreads are primarily led by changes in those of Southeast Asian economies and that changes in Hong Kong's are predicted by all economies except Japan and Pakistan. These findings are consistent with the notion that information about an economy's credit risk diffuses through its major trading partners.

Kazakhstan's leading role is further confirmed in the estimation of the pooled causality regression, in which we restrict the impact of economy *i* to be the same for all other markets. The estimation results show that lagged changes in the CDS spreads of China, Malaysia, and Kazakhstan are significant predictors of the CDS spread changes of the other countries and regions. However, the lead-lag relationship is about three times stronger for Kazakhstan than it is for China and Malaysia.

Overall, the results in this section suggest that lagged changes in Kazakhstan's CDS spreads contain predictive information about CDS spread changes in other economies. In the next section, we conduct a news-diffusion model test to formally examine the degree of information friction across these economies.

**Table 3.** Pairwise Granger causality tests.

| | $\beta^{*}_{i,CHN}$ | $\beta^{*}_{i,HK}$ | $\beta^{*}_{i,INA}$ | $\beta^{*}_{i,JPN}$ | $\beta^{*}_{i,MAS}$ | $\beta^{*}_{i,PAK}$ | $\beta^{*}_{i,PHI}$ | $\beta^{*}_{i,KOR}$ | $\beta^{*}_{i,THA}$ | $\beta^{*}_{i,KAZ}$ |
|---|---|---|---|---|---|---|---|---|---|---|
| CHN | | −0.0251 | −0.0602 | −0.0139 | 0.0248 | −0.0137 | −0.0051 | −0.0345 | −0.0036 | 0.2859 *** |
| | | (−0.5335) | (−0.9627) | (−0.2975) | (0.4477) | (−0.6214) | (−0.0819) | (−0.3622) | (−0.0638) | (5.5023) |
| HK | 0.0732 ** | | 0.0669 ** | 0.0466 | 0.1359 *** | −0.0484 | 0.0704 ** | 0.0681 ** | 0.0635 * | 0.243 *** |
| | (1.6565) | | (1.7137) | (1.0022) | (2.8013) | (−1.7772) | (1.8087) | (1.7087) | (1.5708) | (4.7787) |
| INA | 0.0093 | −0.0534 | | 0.007 | −0.006 | 0.0228 | 0.0274 | −0.0218 | −0.0226 | 0.2829 *** |
| | (0.1471) | (−1.352) | | (0.1836) | (−0.0871) | (1.2316) | (0.47) | (−0.3524) | (−0.344) | (5.6263) |
| JPN | 0.0608 | −0.0273 | 0.0169 | | 0.0895 ** | 0.0373 * | 0.0227 | 0.0471 | 0.051 | 0.1532 *** |
| | (1.294) | (−0.7151) | (0.3636) | | (1.7743) | (1.4268) | (0.4381) | (1.0203) | (0.9768) | (3.3186) |
| MAS | 0.073 | −0.062 | 0.0768 | −0.0109 | | 0.0094 | 0.0802 | 0.0419 | 0.0806 | 0.2784 *** |
| | (0.8895) | (−1.4565) | (0.9197) | (−0.2314) | | (0.5766) | (1.1059) | (0.4764) | (0.9508) | (5.5804) |
| PAK | 0.0313 * | −0.0219 | 0.0312 * | −0.0199 | 0.0168 | | 0.0188 | −0.0082 | 0.0156 | 0.0159 |
| | (1.4626) | (−0.7363) | (1.4975) | (−0.6677) | (0.7263) | | (0.7773) | (−0.3774) | (0.7744) | (0.5277) |
| PHI | 0.0549 | −0.0384 | 0.0968 * | 0.0197 | 0.0757 | 0.0072 | | 0.0498 | 0.0682 | 0.2875 *** |
| | (0.8794) | (−1.0916) | (1.4675) | (0.5056) | (1.1702) | (0.464) | | (0.8512) | (1.0472) | (6.0428) |
| KOR | 0.0708 | −0.0161 | −0.0561 | −0.0107 | 0.0352 | 0.0104 | −0.0114 | | −0.0285 | 0.2659 *** |
| | (0.8467) | (−0.3356) | (−0.9308) | (−0.2011) | (0.5125) | (0.5837) | (−0.2157) | | (−0.5121) | (5.3987) |
| THA | 0.0582 * | −0.0251 | 0.0817 * | −0.0181 | 0.1744 *** | −0.0154 | 0.0965 ** | 0.0537 | | 0.2153 *** |
| | (1.3423) | (−0.7485) | (1.5041) | (−0.5034) | (2.7683) | (−0.9698) | (2.0666) | (1.1106) | | (4.6881) |
| KAZ | 0.0182 | −0.0297 | −0.0079 | 0.0216 | −0.0049 | 0.0279 | −0.0241 | −0.0156 | −0.0076 | |
| | (0.4984) | (−0.788) | (−0.2128) | (0.6491) | (−0.1255) | (1.2475) | (−0.6681) | (−0.4564) | (−0.174) | |
| Average | 0.0500 | −0.0332 | 0.0274 | 0.0024 | 0.0602 | 0.0042 | 0.0306 | 0.0200 | 0.0241 | 0.2253 |
| Pooled | 0.0692 ** | −0.0176 | 0.0514 | 0.0173 | 0.0763 ** | 0.0034 | 0.0531 | 0.0489 | 0.0513 | 0.227 *** |
| | (2.5469) | (−0.7286) | (1.9421) | (0.5893) | (2.4705) | (0.2619) | (1.952) | (1.8297) | (1.5057) | (6.3954) |

Notes: This table reports OLS estimates of $\beta_{i,j}$ (denoted as $\hat{\beta}_{i,j}$) for the predictive regression model: $r_{i,t+1} = \beta_{i,0} + \beta_{i,i}r_{i,t} + \beta_{i,j}r_{j,t} + \beta_{i,S}r_{S,t} + \beta_{i,V}VIX_t + \epsilon_{i,t+1}$ *for* $i \neq j$, where $r_{i,t+1}$ is the log daily change in economy $i$'s CDS spread, $r_{s,t}$ is log return of the S&P 500 index, and $VIX_t$ is the level of the CBOE VIX. Values in parentheses are heteroskedasticity-robust $t$-statistics; $t$-statistics for $\hat{\beta}_{i,S}$ ($\hat{\beta}_{i,V}$) are used to test $H_0: \beta_{i,S} = 0$ against $H_A: \beta_{i,S} < 0$ ($H_0: \beta_{i,V} = 0$ against $H_A: \beta_{i,V} > 0$). Figures in parentheses below the $R^2$ statistics report heteroskedasticity-robust $\chi^2$ statistics for testing $H_0: \beta_{i,S} = \beta_{i,V} = 0$. The "Pooled" estimates impose the restriction that $\beta_{i,j} = \bar{\beta}_j$ for all $i$. *, **, and *** denote significance at the 10%, 5%, and 1% levels, respectively, according to wild bootstrapped $p$-values.

## 5. News-Diffusion Model

Following Rapach et al. (2013 [32]), we estimate a news-diffusion model for CDS spread changes in nine countries and regions using lagged changes in Kazakhstan's CDS spread as predictors:

$$r_{KAZ,t+1} = x'_{KAZ,t}\beta_{KAZ} + u_{KAZ,t+1} \tag{3}$$

$$r_{i,t+1} = x'_{i,t}\beta_i + \theta_{i,KAZ}\lambda_{i,KAZ}u_{KAZ,t+1} + (1 - \theta_{i,KAZ})\lambda_{i,KAZ}u_{KAZ,t} + u_{i,t+1} \tag{4}$$

where $r_{i,t+1}$ is the daily log change in economy $i$'s CDS spread; $\beta_i = (\beta_{i,0}, \beta_{i,S}, \beta_{i,v})^{T}$; $r_{S,t}$ is the log return of the S&P 500 index; $VIX_t$ is the CBOE VIX; $u_{KAZ,t+1}$; $x_{i,t} = [1, r_{S,t}, VIX_t]^{T}$ captures new information reflected in Kazakhstan's CDS spread; and the news-diffusion parameters, $\lambda_{i,KAZ}$ and $\theta_{i,KAZ}$, are constructed to estimate the total impact of a shock from Kazakhstan's CDS spread changes on economy $i$ and the speed of information diffusion from Kazakhstan to economy $i$, respectively. We construct a GMM estimation for the news-diffusion model with relevant orthogonality moment conditions, and Table 4 shows the estimation results. Heteroskedasticity-robust $t$-statistics are shown in parentheses. The $t$-statistics for $\beta_{i,S}$ ($\beta_{i,V}$) are used to test $H_0$:$\beta_{i,S} = 0$ against $H_1$:$\beta_{i,S} < 0$ ($H_0$:$\beta_{i,V} = 0$ against $H_1$:$\beta_{i,V} > 0$). The $t$-statistics for $\widetilde{\theta}_{i,KAZ}$ ($\widetilde{\lambda}_{i,KAZ}$) are used to test $H_0 : \widetilde{\theta}_{i,KAZ} = 1$ against $H_A : \widetilde{\theta}_{i,KAZ} < 1$ ($H_0 : \widetilde{\lambda}_{i,KAZ} = 0$ against $H_A : \widetilde{\lambda}_{i,KAZ} > 0$). The estimates of $\widetilde{\beta}_{i,KAZ}$ are then computed as $\widetilde{\beta}_{i,KAZ} = (1 - \theta_{i,KAZ})\lambda_{i,KAZ}$ to gauge the relative contribution of information frictions to the predictability of lagged changes in Kazakhstan's CDS spreads. The $t$-statistics for $\widetilde{\beta}_{i,KAZ}$ are used to test $H_0 : \widetilde{\beta}_{i,KAZ} = 0$ against $H_A : \widetilde{\beta}_{i,KAZ} > 0$.

The coefficient estimates of $\beta_{i,S}$ and $\beta_{i,V}$ in Table 4 are respectively consistent with those reported in Table 2. We use lagged changes in Kazakhstan's CDS spreads with a lag length of one, and the coefficient of the lagged term is reported in Table 4 as $\widetilde{\lambda}_{i,KAZ}$. Only the U.S. stock market return is negatively and significantly associated with changes in these markets' CDS spreads (except Pakistan). The total impact parameter, $\widetilde{\lambda}_{i,KAZ}$, is significantly positive and consistent with the earlier finding that Kazakhstan plays a leading role. The diffusion speed parameter, $\widetilde{\theta}_{i,KAZ}$, is significantly less than one, suggesting that information friction exists between Kazakhstan's CDS market and other Asian CDS markets. In particular, it implies that it takes more than one day for new information in Kazakhstan's CDS market to be transmitted to other CDS markets in the region. The pooled estimates based on the restriction that all parameters are identical for all $i$ lead to similar conclusions.

For the model specification, the Sargan's $J$-test is conducted, and the corresponding $p$-value for the $J$-statistic is greater than 0.10. Thus, the model fails to reject the overidentification null hypothesis at the 10% significance level, as in the study by Rapach et al. (2013 [32]), to which our news-diffusion model refers. The analysis results in this study must be interpreted in consideration of this.

Comparing the coefficient $\widetilde{\beta}_{i,KAZ}$ in Table 4 with $\hat{\beta}_{i,KAZ}$ in Table 3 allows us to measure the degree to which the predictability of changes in Kazakhstan's CDS spreads can be attributed to information frictions. Similar to the findings by Rapach et al. (2013 [32]) regarding the role of U.S. stock returns in predicting other countries' returns, we find that, on average, 70%–80% of the predictive power of Kazakhstan's CDS spread changes can be attributable to information frictions. This result suggests that the predictive power of lagged changes in Kazakhstan's CDS spreads may be partially attributed, but not limited, to information frictions.

**Table 4.** News-diffusion model parameter estimates.

|  | $\hat{\beta}_{i,S}$ | $\hat{\beta}_{i,V}$ | $\widetilde{\theta}_{i,KAZ}$ | $\widetilde{\lambda}_{i,KAZ}$ | $\widetilde{\beta}_{i,KAZ}$ |
|---|---|---|---|---|---|
| CHN | −1.3096 *** | 0.0041 | 0.644 *** | 0.6102 *** | 0.2172 *** |
|  | (−9.2146) | (0.1807) | (−5.5943) | (8.3096) | (4.2318) |
| HK | −0.9431 *** | 0.0247 | 0.4008 *** | 0.3438 *** | 0.206 *** |
|  | (−5.6546) | (0.8817) | (−4.5896) | (5.2076) | (3.8247) |
| INA | −1.2642 *** | 0.0078 | 0.6536 *** | 0.6108 *** | 0.2116 *** |
|  | (−8.2932) | (0.3435) | (−5.1836) | (8.6721) | (3.986) |
| JPN | −0.9638 *** | 0.0162 | 0.4863 *** | 0.3063 *** | 0.1574 *** |
|  | (−7.3415) | (0.846) | (−5.1089) | (5.9226) | (3.6445) |
| MAS | −1.2058 *** | 0.0103 | 0.5643 *** | 0.5543 *** | 0.2415 *** |
|  | (−9.3148) | (0.4984) | (−7.1116) | (8.0144) | (5.2262) |
| PAK | 0.0178 | 0.0125 | −0.6108 | 0.0145 | 0.0233 |
|  | (0.175) | (0.8662) | (−0.4018) | (0.3218) | (0.6114) |
| PHI | −1.2428 *** | 0.008 | 0.6509 *** | 0.5477 *** | 0.1912 *** |
|  | (−8.5141) | (0.3736) | (−5.3808) | (8.5208) | (4.1307) |
| KOR | −1.4164 *** | 0.005 | 0.6484 *** | 0.5499 *** | 0.1933 *** |
|  | (−9.7681) | (0.2093) | (−5.1779) | (8.1838) | (3.8953) |
| THA | −0.9988 *** | 0.0145 | 0.62 *** | 0.4801 *** | 0.1824 *** |
|  | (−8.9048) | (0.7852) | (−5.6349) | (7.4744) | (3.9836) |
| KAZ | −0.8163 *** | 0.0025 |  |  |  |
|  | (−7.1825) | (0.1268) |  |  |  |
| Pooled | −0.9138 *** | −0.0177 | 0.5801 *** | 0.382 *** | 0.1604 *** |
|  | (−6.4554) | (−0.8811) | (−5.7739) | (4.5785) | (2.9257) |

Notes: This table reports two-step GMM parameter estimates for the news-diffusion models: $r_{KAZ,t+1} = x'_{KAZ,t}\beta_{KAZ} + u_{KAZ,t+1}$ and $r_{i,t+1} = x'_{i,t}\beta_i + \theta_{i,KAZ}\lambda_{i,KAZ}u_{KAZ,t+1} + (1-\theta_{i,KAZ})\lambda_{i,KAZ}u_{KAZ,t} + u_{i,t+1}$, where $r_{i,t+1}$ is the daily log change in economy $i$'s CDS spread; $x_{i,t} = (1, r_{S,t}, VIX_t)'$; $\beta_i = (\beta_{i,0}, \beta_{i,S}, \beta_{i,v})'$; $r_{S,t}$ is the log return of the S&P 500 index, and $VIX_t$ is the level of the CBOE VIX. The GMM estimation is conducted with orthogonality moment conditions that are relevant to the news-diffusion model. Figures in parentheses are heteroskedasticity-robust $t$-statistics. The $t$-statistics for $\beta_{i,S}$ ($\beta_{i,V}$) are used to test $H_0$:$\beta_{i,S} = 0$ against $H_1$:$\beta_{i,|S} < 0$ ($H_0$:$\beta_{i,|V} = 0$ against $H_1$:$\beta_{i,|V} > 0$). The $t$-statistics for $\widetilde{\theta}_{i,KAZ}$ ($\widetilde{\lambda}_{i,KAZ}$) are used to test $H_0 : \widetilde{\theta}_{i,KZA_i} = 1$ against $H_A : \widetilde{\theta}_{i,KAZ} < 1$ ($H_0 : \widetilde{\lambda}_{i,KAZ} = 0$ against $H_A : \widetilde{\lambda}_{i,KAZ} > 0$). The estimates of $\widetilde{\beta}_{i,KAZ}$ are computed as $\widetilde{\beta}_{i,KAZ} = (1 - \theta_{i,KAZ})\lambda_{i,KAZ}$. The $t$-statistics for $\widetilde{\beta}_{i,KAZ}$ are used to test $H_0 : \widetilde{\beta}_{i,KAZ} = 0$ against $H_A : \widetilde{\beta}_{i,KAZ} > 0$. The "Pooled" estimates impose the following homogeneity restrictions: $\beta_{i,S} = \overline{\beta}_S$, $\beta_{i,V} = \overline{\beta}_V$ for all $i$, $\theta_{i,KAZ} = \overline{\theta}_{KAZ}$, $\lambda_{i,KAZ} = \overline{\lambda}_{KAZ}$ for all $i \neq KAZ$. *, **, and *** denote significance at the 10%, 5%, and 1% levels, respectively, according to wild bootstrapped $p$-values.

## 6. Out-of-Sample Evidence

The above in-sample analysis indicates that changes in Kazakhstan's CDS spreads play a leading role in Asian CDS markets. However, the credibility of the in-sample analysis approach has been consistently questioned owing to the potential over-fitting problem. Thus, we also propose an out-of-sample test to check the robustness of the analysis results. In this section, we adopt Campbell and Thompson's (2008 [34]) approach using $R^2$ statistics ($R^2_{OS}$) from out-of-sample tests. Here, $R^2_{OS}$ indicates the mean squared forecast errors' (MSFE) proportional decrease for the benchmark model—the estimation model with a constant term for excess returns—relative to its competing counterpart—the estimation model with lagged log changes in Kazakhstan's CDS spreads. Specifically, the benchmark model and the competing model are given by

$$r_{i,t+1} = \beta_{i,0} + \varepsilon_{i,t+1} \tag{5}$$

and

$$r_{i,t+1} = \beta_{i,0} + \beta_{i,KAZ}r_{KAZ,t} + \varepsilon_{i,t+1}, \tag{6}$$

respectively. Rapach et al. (2013 [32]) adopted this methodology to examine the predictive power of U.S. stock returns, using the out-of-sample approach. In our tests, we use data from 14 February 2011 to 7 January 2013 to estimate the model parameters, and we then use data from 8 January 2013

to 25 November 2014, for the out-of-sample tests. Choosing different widths for the in-sample and out-of-sample windows do not change our results significantly.

The results in Table 5 show that lagged changes in Kazakhstan's CDS spreads can significantly predict CDS spread changes in China, Indonesia, Japan, Malaysia, the Philippines, South Korea, and Thailand outside of the sample (We reject the null hypothesis $R^2_{OS} = 0$ for Hong Kong despite its negative $R^2_{OS}$ based on the MSFE-adjusted statistic). In particular, the information contained in lagged changes in Kazakhstan's CDS spreads increases predictive power for China, Malaysia, the Philippines, and Thailand by more than 5% relative to the benchmark random walk model. The pooled estimate of $R^2_{OS}$ has even greater predictive power.

**Table 5.** Out-of-sample performance.

| $i$ | $R^2_{OS}$ | $R^2_{OS,pooled}$ | $i$ | $R^2_{OS}$ | $R^2_{OS,pooled}$ |
|---|---|---|---|---|---|
| China | 5.0161% *** | 6.0237% *** | Pakistan | −0.1169% | −3.1834% ** |
| | (4.2367) | (4.2435) | | (−0.8863) | (1.7213) |
| Hong Kong | −8.797% *** | −6.567% *** | Philippines | 5.5133% *** | 6.4604% *** |
| | (2.9917) | (2.8767) | | (4.3871) | (4.4028) |
| Indonesia | 3.6263% *** | 5.1942% *** | Korea | 1.1103% *** | 3.2872% *** |
| | (3.9219) | (3.9232) | | (3.4704) | (3.4832) |
| Japan | 1.891% *** | 0.5045% *** | Thailand | 5.6842% *** | 6.088% *** |
| | (4.5437) | (4.5423) | | (4.0056) | (4.038) |
| Malaysia | 5.4405% *** | 7.1952% *** | Average | 2.152% | 2.7781% |
| | (4.842) | (4.8403) | | | |

Notes: This table shows out-of-sample $R^2$ statistics ($R^2_{OS}$) suggested by Campbell and Thompson (2008 [34]). $R^2_{OS}$ indicates the MSFE's proportional decrease for the benchmark model—the estimation model with a constant term for excess returns—relative to its competing counterpart—the estimation model with lagged log changes in Kazakhstan's CDS spreads. The benchmark and competing models are given by $r_{i,t+1} = \beta_{i,0} + \varepsilon_{i,t+1}$ and $r_{i,t+1} = \beta_{i,0} + \beta_{i,KAZ} \, r_{KAZ,t} + \varepsilon_{i,t+1}$, respectively, where $r_{i,t+1}$ is the daily change in the log CDS spreads of economy $i$. The "Pooled" estimates impose the restriction that $\beta_{i,KAZ} = \overline{\beta}_{KAZ}$ for all $i \neq KAZ$ in the competing model. We estimate the out-of-sample forecasts by recursively conducting OLS regressions, employing the available data to formulate daily forecasts. Figures in parentheses below the $R^2_{OS}$ statistics provide the MSFE-adjusted $R^2$ statistics, suggested by Clark and West (2007 [35]), to test $H_0 : R^2_{OS} = 0$ against $H_A : R^2_{OS} > 0$. *, **, and *** denote significance at the 10%, 5%, and 1% levels, respectively, according to wild bootstrapped $p$-values. The in-sample data are from 14 February 2011 to 7 January 2013, and the out-of-sample data are from 8 January 2013 to 25 November 2014.

We also adopt the methods of Goyal and Welch (2003 [36]) and Welch and Goyal (2008 [37]) as an alternative way to look at the out-of-sample predictive power of lagged changes in Kazakhstan's CDS spreads. Figure 2 plots the differences in the cumulative squared forecasting errors between the benchmark random walk model, that uses daily changes in log CDS spread forecasts based on constant expected daily changes in log CDS spreads, and the predictive model, that uses lagged changes in the log of Kazakhstan's CDS spread. The former is given in Equation (5), and the latter is given in Equation (6). We estimate out-of-sample forecasts by recursively conducting OLS regressions, employing the available data to formulate daily forecasts. Positive values indicate better performance of the prediction model against the benchmark model. Figure 2 illustrates that the out-of-sample predictive power of the model including lagged changes in Kazakhstan's CDS spreads is consistently better than that of the benchmark model.

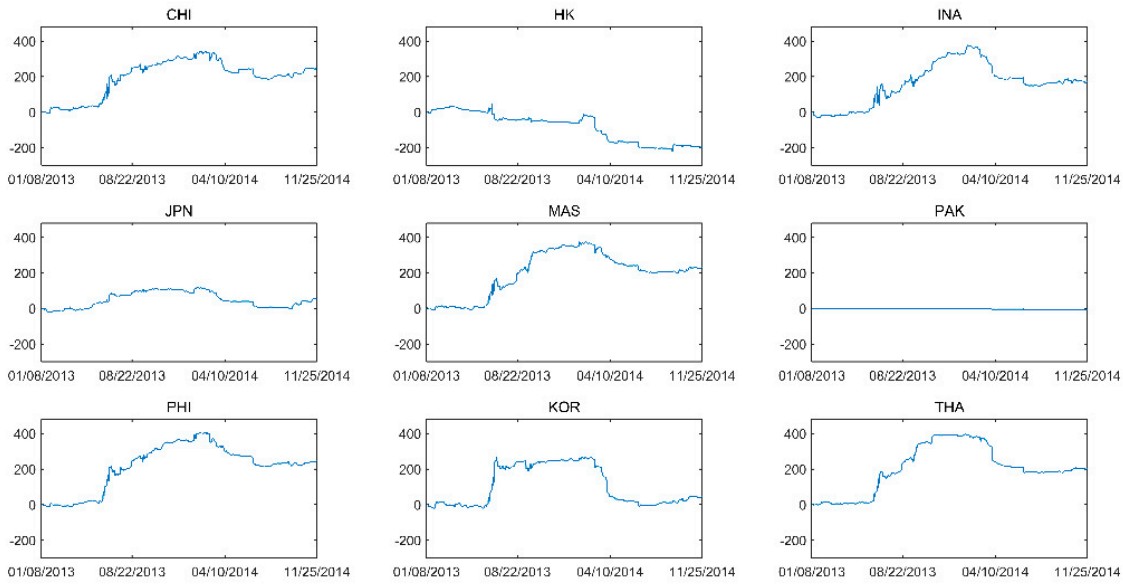

**Figure 2.** Out-of-sample results.

## 7. Conclusions

We investigate the lead-lag relationships among changes in Asian sovereign CDS spreads, which have not been explored previously. Using daily sovereign CDS spreads, we find that Kazakhstan plays a leading role. Using pairwise Granger causality tests and controlling for global impacts, we find that lagged changes in Kazakhstan's sovereign CDS spreads significantly predict changes in other Asian economies' CDS spreads. In order to understand the drivers of Kazakhstan's leading role, we employ a news-diffusion model, as proposed by Rapach et al. (2013 [32]), and find evidence for an information diffusion explanation, that is, information frictions explain a major percentage of the predictive power of lagged changes in Kazakhstan's CDS spreads. Based on these findings, we test the out-of-sample predictability of these lagged changes and show that they significantly predict spread changes in other Asian economies.

The findings above, particularly the out-of-sample test results, have straightforward trading implications for CDS investors and other market participants. The "puzzle" raised by this study regarding the informational and leading role of Kazakhstan's CDS spreads in the region is perhaps more intriguing to academics and policymakers. We offer several potential explanations, including Kazakhstan's credit rating class, its cultural bond with Europe, and its underlying bond being issued by a national gas and oil company instead of its government, but we leave solving this puzzle to the future work.

**Author Contributions:** Proposal & Original Idea, B.G., Q.H., and D.R.; Conceptualization, Q.H., J.L., and D.R.; Modelling, B.G. and D.R.; Methodology, B.G. and J.L.; Data Construction, B.G., and D.R.; Empirical Test, B.G. and J.L.; Validation, Q.H. and J.Y.; Resources, Q.H. and D.R.; Software, B.G.; Literature Review, Q.H. and J.Y.; Economic & Business Implication, Q.H. and D.R.; Writing—Original Draft Preparation, B.G., Q.H., J.L., and D.R.; Writing—Review & Editing, J.Y.; Discussion, B.G., J.L., and J.Y.; Project Administration, B.G. and D.R.; All authors have read and agreed to the published version of the manuscript.

**Funding:** This work was supported by the National Research Foundation of Korea (NRF) grant funded by the Korea government (MSIT; Ministry of Science and ICT) [grant number: 2019R1G1A1100196].

**Acknowledgments:** This paper was presented at the 2015 SSEM conference held in Seoul, which was jointly sponsored by KAIST Graduate School of Finance and Korea Institute of Finance. We are grateful for the valuable comments of Ali M. Kutan, Jangkoo Kang, and two anonymous referees.

**Conflicts of Interest:** The authors declare no conflict of interest.

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
