# Peer review of "Sovereign Credit Spread Spillovers in Asia"

_sustainability, doi:10.3390/su12041472_

Round 1

Reviewer 1 Report

Thank you for giving me the opportunity to read this manuscript ‘Sovereign Credit Spreads Spillover in Asia’. I enjoy reading your paper and I feel the paper makes a good contribution particular to the area of credit exposure activities. Although, the study is significantly motivated by the work of Rapach, Strauss, and Zhou (2013) except for the investigation of different countries, this study extended our knowledge from a different context. However, in its current form, I still feel that some changes need to be incorporated into the paper to bring it to publishable standard and I outline these below;

For GMM estimation, I advise to mention the lag lengths used in all the models. Moreover, in table 4, you need to report the followings for each of the models: Coefficient of the lag of dependent variable, and post-estimation test results i.e. AR (2) and Sargan/ Hansen test statistic.

For literature, you should have very brief discussion about the developing country/emerging country in Asia and why it matters for this study with some supporting evidences. For example,

Aizenman, J., Jinjarak, Y., & Park, D. (2016). Fundamentals and sovereign risk of emerging markets. Pacific Economic Review21(2), 151-177.

Mendy, J., & Rahman, M. (2019). Application of human resource management's universal model: An examination of people versus institutions as barriers of internationalization for SMEs in a small developing country. Thunderbird International Business Review61(2), 363-374.

You can also relate with resilience, for example, Rahman, Mahfuzurand Mendy, John(2019) Evaluating People-related Resilience and Non-Resilience Barriers of SMEs’ Internationalisation: A developing country perspective. International Journal of Organizational Analysis, 27 (2). pp. 225-240.

Author Response

Author Responses for the Referee Comments

Manuscript ID: sustainability-706948

Type of Manuscript: Article

Title: Sovereign Credit Spreads Spillover in Asia

Overview Comment from Referee #1

Thank you for giving me the opportunity to read this manuscript ‘Sovereign Credit Spreads Spillover in Asia’. I enjoy reading your paper and I feel the paper makes a good contribution particular to the area of credit exposure activities. Although, the study is significantly motivated by the work of Rapach, Strauss, and Zhou (2013) except for the investigation of different countries, this study extended our knowledge from a different context. However, in its current form, I still feel that some changes need to be incorporated into the paper to bring it to publishable standard and I outline these below.

Overall Answer for Referee #1

We appreciate the referee for reading our paper ‘Sovereign Credit Spreads Spillover in Asia,’ which examines the relationships of sovereign CDS spreads among Asian economies, and for providing helpful comments for us to revise. We agree with your comments and believe that this paper is much improved with the addition of the advised materials, including the model specification, discussion regarding emerging market features, and the importance to shed light on theis specific type of economies.

Referee #1, Comment 1)

For GMM estimation, I advise to mention the lag lengths used in all the models. Moreover, in table 4, you need to report the followings for each of the models: Coefficient of the lag of dependent variable, and post-estimation test results i.e. AR (2) and Sargan/ Hansen test statistic

Author Answer for Comment 1 of Referee #1

We agree with your comment that the model structure of the GMM estimation needs to be more clearly specified. The lagged CDS spread changes of Kazakhstan is used in the model estimation in Table 4, and the lag length is one. The coefficient for the lagged term is reported in the column  of Table 4. We conduct the Sargan’s J test for the overidentification restriction of this model, and the test reports the J-statistic of 4.48e-05 and corresponding p-value of 0.99. Hence, model does not reject the overidentification null hypothesis, which is also the case for Rapach, Strauss, and Zhou (2013), which the motivative study for the news-diffusion model used in our study. Therefore, taking your advice, we add the following as a footnote to provide further information to readers about the specification of the estimated model:

(p. 14 of 20)

3 We use the lagged CDS spread changes of Kazakhstan with the lag length of one, and the coefficient of the lagged term is reported in Table 4 as . For the model specification, the Sargan’s J test has been conducted and the corresponding p-value for the J-statistic is greater than 0.10. Thus, the model fails to reject the overidentification null hypothesis at 10% significance level as the study by Rapach, Strauss, and Zhou (2013), to which our news-diffusion model refers, does. The interpretation of the analysis needs to be done in consideration of this.”

Referee #1, Comment 2)

For literature, you should have very brief discussion about the developing country/emerging country in Asia and why it matters for this study with some supporting evidences. For example,

Aizenman, J., Jinjarak, Y., & Park, D. (2016). Fundamentals and sovereign risk of emerging markets. Pacific Economic Review, 21(2), 151-177.

Mendy, J., & Rahman, M. (2019). Application of human resource management's universal model: An examination of people versus institutions as barriers of internationalization for SMEs in a small developing country. Thunderbird International Business Review, 61(2), 363-374.

You can also relate with resilience, for example, Rahman, Mahfuzurand Mendy, John(2019) Evaluating People-related Resilience and Non-Resilience Barriers of SMEs’ Internationalisation: A developing country perspective. International Journal of Organizational Analysis, 27 (2). pp. 225-240.

Author Answer for Comment 2 of Referee #1

We thank you for your suggestion to further develop the literature section by illustrating the significance of studying emerging markets and to relate our study to the context of resilience. We believe this suggestion is valid, and it is important to explain reasons why we specifically focus on relationships among sovereign CDS spreads of Asian economies, or more generally, emerging economies.

Previous studies provide firm evidence for the uniqueness of emerging markets. For example, Alon and Rottig (2013) address the potential difference of emerging markets in comparison with advance markets and explain that studying these markets can contribute to conventional theories by enhancing their generality or further adjusting them to explain new research environments (i.e., emerging markets). There are studies that claim emerging economies differ from their developed counterparts, implying the necessity to explore these markets (Aizenman, Jinjarak, and Park, 2016; Mendy and Rahman, 2019). In addition, Aizenman, Jinjarak, and Park (2016) and Rahman and Mendy (2019) each emphasize the strong resilience of emerging economies and investigate the importance of resilience-related factors within an emerging market, respectively, both implying autonomous features of these markets. At the same time, there are also studies indicating that these markets are not entirely decoupled from advanced markets. For example, Bowman, Londono, and Sapriza (2015) show that the U.S. monetary policies significantly influence sovereign bond yields of emerging markets; Fink and Schüler (2015) show that the U.S. financial shocks significantly influence macroeconomic dynamics of emerging markets. Song, Park, and Ryu (2018) also investigate the dynamic conditional relationship between advanced and emerging markets and find that the advanced market’s financial factors significantly affect the dynamic relationship. Findings of these studies imply that emerging markets possess unique and independent features, and yet are highly interactive with their advanced counterparts, indicating the importance to investigate these markets.

Acknowledging such features of emerging markets, we add following paragraphs to the introduction of our paper:

(p. 3-4 of 20)

“While investigating the lead-lag relationships between countries, it is important that this study particularly focuses on the relationships among emerging economies. Previous studies on relevant topics mostly focus on the advanced economies or the relationship between advanced and developing economies. This study uniquely investigates the CDS spreads’ spillover among Asian emerging countries since these markets provide different research context. Many previous studies emphasize the importance of economic analyses upon emerging economies and the uniqueness of their market structures varying from advanced ones. Alon and Rottig (2013) suggest that emerging markets are different from its advanced counterparts in that they possess different types of political, economic structures and cultural conventions. They suggest that different economic contexts of these developing economies provide new environments to study conventional ideas or theories, leading to further generalization or adjustment of traditional ones. Aizenman, Jinjarak, and Park (2016) assert that emerging markets exhibit much faster growth compared to advanced markets, possibly implying the two-speed world economy theory. Mendy and Rahman (2019) study the internationalization of small to medium-sized enterprises (SMEs) in an emerging market, considering potential barriers regarding the people-relevant—such as language difference, scarcity of experienced workers, and so on—and institution-relevant factors—such as how instable political activities are, how cumbersome the legal procedures for internationalization are, and so on. They assert that emerging markets are vastly different from advanced ones particularly regarding their people and government-related backgrounds.

However, emerging markets are not completely detached form the world economy and rather exhibits significantly interactive, and yet autonomous, behaviors in response to the sovereign and financial activities of advanced markets. There are numbers of studies illustrating the connection between advanced and emerging markets. For example, Bowman, Londono, and Sapriza (2015) reveal that the U.S. monetary policies significantly influence yields of sovereign bonds of emerging markets. Similarly, Fink and Schüler (2015) find that shocks upon the U.S. financial factors, such as its GDP, CPI, and so on, play as significant determinants of emerging markets macroeconomic dynamics. In addition, Song, Park, and Ryu (2018) investigate the dynamic relationship between advanced and emerging markets and find that the financial variables—especially, the U.S. S&P 500 returns and VIX—better explain the relationship than macroeconomic variables, such as the exchange rate, interest rate, and so on. At the same time, there are studies that address independent aspects of emerging markets, specifically about their resilience to the international economic status. Aizenman, Jinjarak, and Park (2016) claim that emerging markets displayed greater resilience during the global financial crisis period around 2008, particularly in comparison with advanced economies such as the US and Eurozone. Also, Rahman and Mendy (2019) investigate SMEs in an emerging market and find that emerging market SMEs internationalizations are significantly affected by resilience-related factors, especially the language discrepancy barrier. Therefore, in line with the literature, we believe that it is academically necessary and timely to investigate the CDS spreads relationships under the context of emerging markets.”

Comment from Referee #2

This paper provides an analysis of sovereign credit default swap spread spillovers in Asia. The paper reaches an interesting finding on the key role played by Kazakhstan in spillovers and the introduction currently does a very good at setting the context for why Kazakhstan might play this role there is some additional literature that supports this view and could also be discussed. Perhaps you could consider adding discussion of Bouri et al (2017) and Li et al (2019) in the introduction.

References

Bouri E., de Boyrie M.E., Pavlova I. (2017), Volatility transmission from commodity markets to sovereign CDS spreads in emerging and frontier countries, International Review of Financial Analysis, 49, pp. 155-165.

Li, H., Semeyutin, A., Lau, C., Gozgor, G. (2019), The relationship between oil and financial markets in emerging economies: The significant role of Kazakhstan as the oil exporting country, Finance Research Letters, forthcoming.

Author Answer for Comment of Referee #2

We appreciate the referee for reading our paper ‘Sovereign Credit Spreads Spillover in Asia,’ which examines the relationships of sovereign CDS spreads among the Asian economies, and for helpful suggestions regarding the introduction and literature of this paper. Indeed, this paper investigates sovereign CDS spreads’ spillover within the Asian region, and as you have pointed out, it could prominently improve this study to include more detailed literature regarding CDS spreads in emerging markets and the regional spillover in the Asian region. Thus, we add the references suggested and revised the introduction of our paper as follows:

(p. 2 of 17)

“A credit default swap (CDS) is a type of protection guarantee or contract where its buyer provides a payment to its seller, which is commonly known as the CDS spread, and in its return, gets compensated in the case of actual default (Bhanot and Guo, 2012; Guo, 2016; Guo and Newton, 2013; Park, Kutan, and Ryu, 2019). There are studies that suggest domestic determinants of the sovereign CDS spreads. For example, Bouri, de Boyrie, and Pavlova (2017), where they investigate six frontier economies (including Croatia, Kazakhstan, and so on) and 17 developing economies (including Brazil, Chile, and so on), and find that the volatility of commodity sectors, especially energy commodities, significantly affect the CDS spread of the country. Nevertheless, the majority of variation in sovereign CDS spreads is accounted for by global factors, as has been clearly shown in numerous papers. For example, Pan and Singleton (2008) apply a single latent factor model on a full term structure of CDS spreads with maturities of one, two, three, five, and 10 years for Mexico, Turkey, and Korea. They find the credit spreads for these three countries are strongly related to the Chicago Board Options Exchange (CBOE) market Volatility Index (VIX). Longstaff, Pan, Pedersen, and Singleton (2011) use five-year CDS data for 26 countries and, by regressing the CDS spreads on both global and local variables, find that sovereign credit spreads are driven more by the US macroeconomic factors than by a country’s local market. Li, Semeyutin, Lau, and Gozgor (forthcoming) investigate how volatiility spillovers between Brent oil price, VIX and sovereign CDS spreads of some oil-importing (South Africa and Turkey) and exporting countries (Kazakhstan and Russia) and Kazakhstan’s role among them. They find that the five-year sovereign CDS of Kazahkstan can be utilized for portfolio diversification, since it exhibits signifciant correlation with those of other countries, and further propose potentail expansion of the research by analyzing the determinants CDS spreads. Other influential studies on sovereign contagion effects include those by Ang and Longstaff (2013), Arora and Cerisola (2001), Benzoni, Collin-Dufresne, Goldstein, and Helwege (2014), Dailami, Masson, and Padou (2008), Gande and Parsley (2005), Geyer, Kossmeier, and Pichler (2004), Mauro, Nathan, and Yishay (2002),  Remolona, Scatigna, and Wu (2008), and Weigel and Gemmill (2006).”

Additional References:

Aizenman, J., Jinjarak, Y., and Park, D. (2016). Fundamentals and sovereign risk of emerging markets. Pacific Economic Review, 21(2): 151-177.

Alon, I., and Rottig, D. (2013). Entrepreneurship in emerging markets: New insights and directions for future research. Thunderbird International Business Review, 55(5): 487-492.

Bouri, E., de Boyrie, M.E., and Pavlova, I. (2017). Volatility transmission from commodity markets to sovereign CDS spreads in emerging and frontier countries. International Review of Financial Analysis, 49: 155-165.

Bowman, D., Londono, J.M., and Sapriza, H. (2015). US unconventional monetary policy and transmission to emerging market economies. Journal of International Money and Finance, 55: 27-59.

Fink, F., and Schüler, Y. S. (2015). The transmission of US systemic financial stress: Evidence for emerging market economies. Journal of International Money and Finance, 55: 6-26.

Li, H., Semeyutin, A., Lau, C K.M., and Gozgor, G. (forthcoming). The relationship between oil and financial markets in emerging economies: The significant role of Kazakhstan as the oil exporting country. Finance Research Letters.

Mendy, J., and Rahman, M. (2019). Application of human resource management's universal model: An examination of people versus institutions as barriers of internationalization for SMEs in a small developing country. Thunderbird International Business Review, 61(2): 363-374.

Rahman, M., and Mendy, J. (2019). Evaluating people-related resilience and non-resilience barriers of SMEs’ internationalisation. International Journal of Organizational Analysis, 27(2): 225-240.

Song, W., Park, S.Y., and Ryu, D. (2018). Dynamic conditional relationships between developed and emerging markets. Physica A: Statistical Mechanics and its Applications, 507: 534-543.

Reviewer 2 Report

This paper provides an analysis of sovereign credit default swap spread spillovers in Asia. The paper reaches an interesting finding on the key role played by Kazakhstan in spillovers and the introduction currently does a very good at setting the context for why Kazakhstan might play this role there is some additional literature that supports this view and could also be discussed. Perhaps you could consider adding discussion of Bouri et al (2017) and Li et al (2019) in the introduction.

References

Bouri E., de Boyrie M.E., Pavlova I. (2017), Volatility transmission from commodity markets to sovereign CDS spreads in emerging and frontier countries, International Review of Financial Analysis, 49, pp. 155-165.

Li, H., Semeyutin, A., Lau, C., Gozgor, G. (2019), The relationship between oil and financial markets in emerging economies: The significant role of Kazakhstan as the oil exporting country, Finance Research Letters, forthcoming.

Author Response

(The authors gave the same response as above.)
